# BoundRL: Efficient Structured Text Segmentation through Reinforced Boundary Generation

## Abstract

As structured texts become increasingly complex across diverse domains – from technical reports to generative AI prompts – the need for text segmentation into semantically meaningful components becomes critical. Such texts often contain elements beyond plain language, including tables, code snippets, and placeholders, which conventional sentence- or paragraph-level segmentation methods cannot handle effectively. To address this challenge, we propose `BoundRL`, a novel and efficient approach that jointly performs token-level text segmentation and label prediction for long structured texts. Instead of generating complete contents for each segment, it generates only a sequence of starting tokens and reconstructs the complete contents by locating these tokens within the original texts, thereby reducing inference costs by orders of magnitude and minimizing hallucination. To adapt the model for the output format, `BoundRL` performs reinforcement learning with verifiable rewards (RLVR) with a specifically designed reward that jointly optimizes document reconstruction fidelity and semantic alignment. To mitigate entropy collapse, it further constructs intermediate candidates by systematically perturbing a fraction of generated sequences of segments to create stepping stones toward higher-quality solutions. To demonstrate `BoundRL`'s effectiveness on particularly challenging structured texts, we focus evaluation on complex prompts used for LLM applications. Experiments show that `BoundRL` enables small language models (1.7B parameters) to outperform few-shot prompting of much larger models. Moreover, RLVR with our designed reward yields significant improvements over supervised fine-tuning, and incorporating intermediate candidates further improves both performance and generalization.

## 1 Introduction

Text segmentation is the task of dividing a text into coherent segments, each covering a distinct topic (Hearst, 1994). Beyond identifying segment boundaries, some approaches also predict the topic of each segment (Arnold et al., 2019b; Barrow et al., 2020). These segments can help readers to better understand the structure of long texts (Jeoung et al., 2025), QA systems to retrieve more relevant contexts (Tiedemann & Mur, 2008; Wang et al., 2025), and summarization system to summarize long documents (Moro & Ragazzi, 2022).

Most previous works frame text segmentation as sequence labeling (Hearst, 1994) or boundary classification (Lukasik et al., 2020) on the sentence or paragraph level. However, these methods assume that texts can be cleanly divided into sentences or paragraphs in advance, which does not hold for many real-world structured texts, such as technical reports or prompts for large language models (LLMs). Such texts often contain tables, code snippets, and placeholders; which do not conform to conventional sentence or paragraph structures, making existing formulations insufficient for increasingly complex structured texts. A natural solution is to modify these methods to perform text segmentation at the token level. However, sequence labeling on the token level tends to generate very fragmented segments, while boundary classification requires an impractically large number of classifications for each token. Recent work addresses token-level segmentation by instructing LLMs to generate the full text of each segment (Schnabel & Neville, 2024; Jeoung et al., 2025). However,

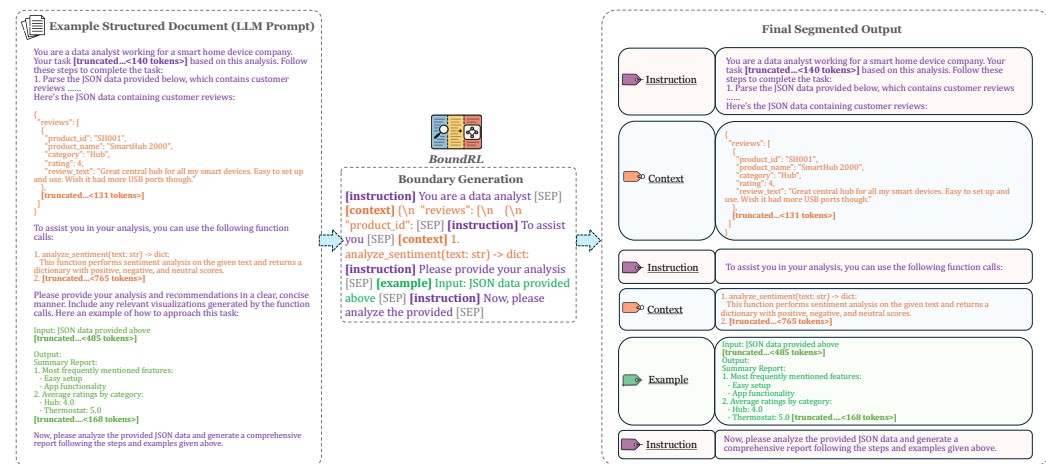

Figure 1: Efficient output pattern used by BoundRL. Instead of generating complete segment text, it only generates starting tokens for each segment and then reconstructs complete segments by locating these tokens in the original text, which reduces inference costs and risks for hallucinations.

these methods incur prohibitive inference costs for long texts due to the necessity of regenerating the entire input, and they are prone to hallucinations (Wang et al., 2024).

We introduce BoundRL, a novel approach that jointly performs token-level text segmentation and label prediction for long structured texts, which we term *structured text segmentation*. As shown in Fig. 1, BoundRL reformulates text segmentation as *boundary generation* and only generates a sequence of starting tokens and label for each segment, then reconstructs full segments by locating them in the input. Unlike approaches that generate every segment in full, this formulation reduces inference costs by orders of magnitude – from $O(|d|)$ to $O(n)$ tokens, where $|d|$ is the document length, $n$ is the average number of segments per document, and $n$ is generally much smaller than $|d|$. It also mitigates inherent hallucination risks during text generation.

Boundary generation training presents unique optimization challenges. Supervised fine-tuning (SFT) can mistakenly penalize starting tokens that correspond to the right boundary positions and provides insufficient penalties for minor token mismatches that cause failures in locating starting tokens. BoundRL addresses this by reinforcement learning with verifiable rewards (RLVR) (Shao et al., 2024), optimizing a reward function with two complementary dimensions: *reconstruction fidelity*, which measures whether the text can be fully recovered from generated segments, and *semantic alignment* which evaluates agreements between generated segments and annotated segments.

However, RLVR can suffer from entropy collapse (Cui et al., 2025), where generated sequences of segments become trapped in narrow, low-reward regions during rollout. Although annotated sequence of segments provides high-reward examples, they are often too distant from the model's current generation distribution to enable effective learning. To mitigate this, BoundRL constructs intermediate candidates by perturbing a fraction of generated sequences of segments through boundary adjustments and label modifications as shown in Fig. 2, creating stepping stones that bridge the gap between current generations and optimal solutions. This approach is particularly effective for our reward function due to its dense, continuous nature.

To validate BoundRL on particularly challenging structured texts, we construct StructSeg, a comprehensive dataset for structured text segmentation with 15.3K annotations encompassing synthetic prompts and prompts from LangSmith [1] with text and label of each segment. Our evaluation focuses on prompts for LLMs due to their extreme structural complexity – dense mixtures of natural language instructions, code snippets, JSON formatting, and contextual data within compact formats, making them unsuitable for sentence- or paragraph-level segmentation.

Our experiments show that relatively small models (1.7B-4B parameters) trained with BoundRL outperforms few-shot prompting using much larger models (Claude-4 Sonnet (Anthropic, 2025)). Moreover, RLVR with our designed reward function brings significant performance and

---

[1]https://smith.langchain.com/hub/

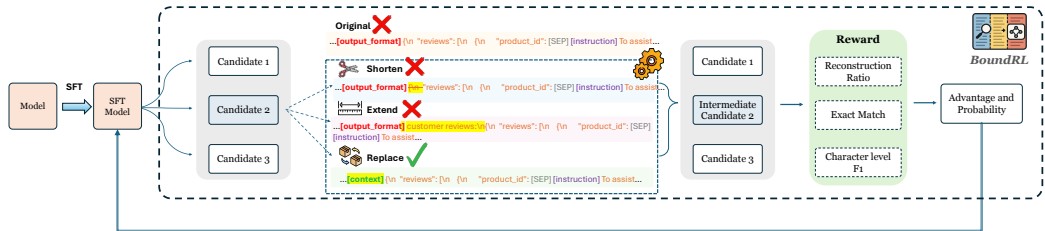

Figure 2: RLVR workflow of `BoundRL` showing the dual-objective reward function and intermediate candidate construction. The reward function combines reconstruction ratio for measuring reconstruction fidelity with exact match and character-level F1 scores for measuring semantic alignment. To mitigate entropy collapse during rollout, `BoundRL` constructs intermediate candidates by perturbing generated segments through boundary adjustments and label modifications.

generalization improvement over SFT and the intermediate candidates can further improve the performance of RLVR.

Our contributions are four-fold:

- A novel boundary-generation approach for structured text segmentation that reduces inference costs from $O(|d|)$ to $O(n)$ tokens while reducing hallucination risks;
- A specifically designed reward function that jointly optimizes reconstruction fidelity and semantic alignment for RLVR;
- Intermediate candidate construction that mitigates entropy collapse in RLVR through boundary adjustments and label modifications;
- Comprehensive evaluation on `StructSeg` benchmark demonstrating `BoundRL`'s significant efficiency and accuracy improvements, with strong out-of-domain generalization.

## 2 RELATED WORK

**Text segmentation** Text segmentation aims to divide a text into coherent segments, where each segment encompasses a distinct semantic unit or topic (Hearst, 1994). The task has been studied across various domains, such as regular documents (Koshorek et al., 2018b) and dialogues (Xing & Carenini, 2021). Arnold et al. (2019b); Barrow et al. (2020) extend the task by jointly modeling segmentation and topic classification. In supervised settings, segmentation is often framed as sequence labeling, where each sentence is labeled as a boundary or not (Koshorek et al., 2018a; Li et al., 2018). Other works frame text segmentation as boundary classification task predicting whether a sentence is a boundary based on its surrounding context (Lukasik et al., 2020). More recently, (Inan et al., 2022; Duarte et al., 2024) frame the text segmentation as a generation task by generating the starting sentence or paragraph indices of each segment. However, these approaches perform segmentation on the sentence or paragraph level, which face limitations when applied to structured texts with code snippets, structured data formats (JSON, XML) that do not follow boundary patterns of traditional sentence or paragraph. As such, `BoundRL` performs text segmentation on the token level, making it applicable to structured texts with diverse elements other than plain texts.

**Text Segmentation Applications** Text segmentation can benefit many downstream applications. (Mao et al., 2025; Jeoung et al., 2025) propose taxonomies for segmenting LLM prompts and examine how different component orders affect final performance. In retrieval-based question-answering, Tiedemann & Mur (2008); Duarte et al. (2024); Wang et al. (2025) apply text segmentation before retrieval so that more relevant text units can be used for the following QA task. In summarization, Moro & Ragazzi (2022) segment long documents into shorter chunks and summarize each chunk separately, while Cho et al. (2022) find that jointly fine-tune a model to perform summarization and text segmentation can further improve the summarization performance.

**Reinforcement Learning with Verifiable Reward** In comparison with RLHF, which relies on a separate reward model to assign rewards (Ouyang et al., 2022), RLVR (Shao et al., 2024) uses a rule-based reward function. This design makes RLVR particularly efficient for structured text segmentation, as rewards can be directly computed by comparing generated and annotated segments.

However, candidates generated during rollout suffer from being trapped in narrow, low-reward regions; also known as entropy collapse (Cui et al., 2025). To mitigate such issue, Zhang et al. (2025) propose to generate candidates conditioned on the prefix of reference data with varying lengths. However, it requires generating as many candidates as there are segments in the input text, leading to high training costs for structured text segmentation. Yan et al. (2025); Dong et al. (2025) additionally include reference candidates generated by a reference model, which can be too distant from the model's current generation distribution for effective learning. In contrast, `BoundRL` proposes to generate intermediate candidates which are closer to the current distribution of generation.

## 3 PROBLEM STATEMENT

Structured text segmentation takes a structured text as input and generates a list of segments:

$$[(l_1, t_1), ..., (l_n, t_n)] = f(d), \ s.t. \ l_i \neq l_{i-1}, t_i \cap t_{i-1} = \emptyset, \ \forall i = 2, ..., n \quad (1)$$

where $f$ denotes the segmentation system, $d$ denotes the input structured text, $t_i$ denotes the text of the $i$-th segment and $l_i \in L$ denotes the semantic label of the $i$-th segment. $L$ denotes the set of potential labels for segments, which varies by domains. For our case study on prompts, labels include 'instruction', 'example', 'context', 'question', and 'output format' as shown in Table 5.

## 4 METHOD

The training process of `BoundRL` consists of two stages: SFT followed by RLVR. Sec. 4.1 describes how to adapt LLMs to an efficient output pattern for structured text segmentation via SFT. Sec.4.2 describes the reward design of RLVR. Sec. 4.3 describes the construction of intermediate candidates for the rollout stage of RLVR.

### 4.1 EFFICIENT OUTPUT PATTERN FOR STRUCTURED TEXT SEGMENTATION

To enable efficient structured text segmentation, `BoundRL` formulates the task as boundary generation. Instead of regenerating full segment text, it produces only a sequence of starting tokens and a label for each segment, then reconstructs the segments from the input as shown in Fig. 1. Specifically, given an input structured text $d$, `BoundRL` tunes the LLM to generate a sequence of starting tokens $s_i$ and a label $l_i$ for each segment: $[(\hat{l}_i, \hat{s}_i)_{i=1:n}] = LLM(d)$. To adapt LLMs for boundary generation, `BoundRL` transforms the annotated text of each segment $t_i$ into corresponding starting token sequences $s_i$. Sequence lengths of each segment are randomly sampled while ensuring that each sequence of starting tokens is unique among each other to improve model robustness. The LLMs are then fine-tuned using SFT on these transformed starting tokens sequences and their corresponding labels.

`BoundRL` then reconstructs the text of each segment $\hat{t}_i$ using the position of each sequence of starting tokens $\hat{s}_i$ in the input $d$. The reconstruction process operates iteratively. Specifically, for the first sequence of starting tokens $\hat{s}_1$, `BoundRL` locates it as its leftmost occurrence in the input $d$. For each subsequent sequence $\hat{s}_i$, `BoundRL` locates it as its leftmost occurrence in document $d$ after the position of its previous segment to preserve ordering. The text of the $i$-th segment $\hat{t}_i$ is then extracted as the text span between sequences of starting tokens $\hat{s}_i$ and $\hat{s}_{i+1}$. If the positions of either $\hat{s}_i$ or $\hat{s}_{i+1}$ cannot be found, the $i$-th segment will be discarded. Compared with regenerating full text of each segment, `BoundRL` reduces hallucination risks inherent in text generation while reducing inference cost from $O(|d|)$ to $O(n)$ tokens, where $|d| \gg n$ denotes the document length.

### 4.2 REWARD DESIGN OF REINFORCEMENT LEARNING

In this section, we describe the reward function used by `BoundRL` for RLVR. The reward function has two dimensions: *reconstruction fidelity*, which measures whether the input can be fully recovered from generated segments, and *semantic alignment* which evaluates agreements between generated segments and annotated segments.

To measure reconstruction fidelity, `BoundRL` uses the reconstruction ratio $\rho_{\text{rec}}$. The metric is calculated as the proportion of the input text $d$ that can be successfully reconstructed from the texts of

generated segments $\hat{t}_i$:

$$\rho_{\text{rec}}([(\hat{l}_i, \hat{t}_i)_{i=1:n}], d) = \frac{\sum_{i=1}^n |\hat{t}_i|}{|d|} \tag{2}$$

where $|*|$ denotes the length of a text in character. The reconstruction ratio $\rho_{\text{rec}}(*)$ ranges from zero to one, with higher values indicating more complete reconstruction of the input structured text $d$.

To measure semantic alignment, `BoundRL` uses two metrics. The first metric is the F-1 score of exact match $\text{EM}([(\hat{l}_i, \hat{t}_i)_{i=1:n}], [(l_i, t_i)_{i=1:n}])$ between generated segments $[(\hat{l}_i, \hat{t}_i)_{i=1:n}]$ and annotated segments $[(l_i, t_i)_{i=1:n}]$ (Tjong Kim Sang & De Meulder, 2003). A generated segment $(\hat{l}_i, \hat{t}_i)$ is considered an exact match to an annotated segment $(l_j, t_j)$ if their texts and labels are the same. The F1 score is then computed as the harmonic mean of precision (the fraction of generated segments with at least one exact match) and recall (the fraction of annotated segments with at least one exact match). However, the F-1 score of exact match $\text{EM}(*)$ can be too strict since minor token-level differences between segment texts can lead to a mismatch. Therefore, `BoundRL` additionally uses the character-level F-1 score $\text{F1}_{\text{char}}([(\hat{l}_i, \hat{d}_i)_{i=1:n}], [(l_i, d_i)_{i=1:n}])$ motivated by part-of-speech (POS) tagging (Marcus et al., 1993). The metric treats structured text segmentation as a character-level labeling task, where each character in the input text $d$ is assigned a label from the set of potential labels $L$, based on the segment it belongs to. Specifically, all characters in $\hat{t}_i$ are assigned the label $\hat{l}_i$, and likewise for annotated segments. The metric is then calculated as the weighted F-1 score between character-level labels from generated segments and those from annotated segments. Both the F-1 score of exact match $\text{EM}(*)$ and the character-level F-1 score $\text{F1}_{\text{char}}(*)$ range from zero to one, with higher values indicating better alignment between generated segments and annotated segments.

The final reward $r(*)$ for the generated segments $[(\hat{l}_i, \hat{d}_i)_{i=1:n}]$ is calculated using both dimensions:

$$r([(\hat{l}_i, \hat{t}_i)_{i=1:n}]) = \rho_{\text{rec}}([(\hat{l}_i, \hat{t}_i)_{i=1:n}]) \times \frac{\text{EM}([(\hat{l}_i, \hat{t}_i)_{i=1:n}]) + \text{F1}_{\text{char}}([(\hat{l}_i, \hat{t}_i)_{i=1:n}])}{2} \tag{3}$$

For simplicity, the equation omits the input text $d$ and the annotated segments $[(l_i, t_i)_{i=1:n}]$. The reward encourages generated segments to accurately reproduce starting tokens for complete reconstruction while getting close to annotated segments for high-quality segmentation.

## 4.3 CONSTRUCTION OF INTERMEDIATE CANDIDATE

In this section, we describe how `BoundRL` constructs and incorporate intermediate candidates during the rollout stage of RLVR to mitigate entropy collapse. Specifically, `BoundRL` constructs intermediate candidates by perturbing generated candidate segmentations and selectively replaces the originally generated candidate segmenetations with the intermediate candidates for the training.

An effective intermediate candidate should be in the middle between generated segments and annotated segments to provide meaningful guidance and remain learnable. To construct such intermediate candidates, `BoundRL` first generates $m$ candidate segmentations for an input text $d$ following standard RLVR practice, denoted as $[(\hat{l}_i, \hat{t}_i)_{i=1:n}]_j$ for $j = 1, \ldots, m$. These candidate segmentations are ordered by descending reward $r([(\hat{l}_i, \hat{t}_i)_{i=1:n}]_j)$. `BoundRL` then perturbs the candidate segmentation with the medium-level reward: $[(\hat{l}_i, \hat{t}_i)_{i=1:n}]_{\frac{m}{2}}$. As shown in Fig. 2, three types of perturbations are considered for each segment: (1) shortening the text $\hat{t}_i$ by truncating one word from either side, (2) extending the text $\hat{t}_i$ by including additional one word from either side, or (3) replacing the label $\hat{l}_i$ with an alternative from the set of potential labels $L$, excluding labels already assigned to neighboring segments. To shorten or extend the text of a segment $\hat{t}_i$, `BoundRL` modifies the starting token sequences $\hat{s}_i$ or $\hat{s}_{i+1}$ accordingly. Applying a single perturbation to each segment creates a pool of potential intermediate candidates, each differing from the original by exactly one perturbation. The potential intermediate candidate with the highest reward $r(*)$ is selected as final the intermediate candidate, denoted as $[(\hat{l}_i, \hat{d}_i)_{i=1:n}]_{\frac{m}{2}}^{pert}$.

To avoid performance degradation from off-policy intermediate candidates, `BoundRL` employs a selective replacement strategy. In each training batch, `BoundRL` replaces the original candidate segmentations with the medium-level reward with the intermediate candidate for at most $k$ input texts. Replacement is allowed only when the intermediate candidate achieves a positive reward

gain over the original: $r([(\hat{l}i, \hat{d}i)_{i=1:n}]_{\frac{m}{2}}^{pert}) - r([(\hat{l}i, \hat{d}i)_{i=1:n}]_{\frac{m}{2}}) > 0$. If more than $k$ input texts satisfy this criterion, `BoundRL` selects the top-$k$ with the largest gains. For the selected cases, `BoundRL` uses the intermediate candidate together with the remaining $m - 1$ generated candidate segmentations in the following training; otherwise, all $m$ original candidates are used. The strategy preserves training stability while incorporating the most beneficial refinements.

# 5 EXPERIMENT SETUPS

## 5.1 STRUCTSEG

In this section, we describe the `StructSeg`, used as a case study of structured text segmentation. It contains synthetic prompts and real-world prompts. The synthetic prompts are generated using Claude 3.5 Sonnet (Anthropic, 2024), following a strategy that balances diversity and complexity. To ensure diversity, we implement a multi-faceted sampling strategy that draws from varied prompt types (system prompt, user prompt, combined), prompt modes (prompt template, full prompt, hybrid), and task types (e.g. classification, summarization), placeholder formats (e.g., {context} or {{context}}), the number of examples, prompt lengths, format types, writing styles, and levels of details. For complexity, we encourage the generated prompts to include varied structural elements, including nested json, code snippets, and placeholders, which make the prompts unsuitable for sentence- or paragraph-level segmentation. The dataset includes multiple lengthy prompts, with some containing around 2,000 words. More details of synthetic prompts are in App. A.6. For real-world prompts, we collect them from Langchain-hub [2].

After collecting both synthetic and real-world prompts, we train a group of highly experienced human annotators to perform prompt segmentation and labeling. Each segment is assigned one of five labels: 'instruction', 'example', 'context', 'question', or 'output format' (Table 5). This simplified taxonomy captures the common structural elements of complex prompts (Mao et al., 2025; Jeoung et al., 2025). To ensure high annotation quality, we develop step-by-step labeling instructions for annotators. Annotators first decompose each prompt into mutually exclusive, non-overlapping segments. The segmentation should also be lossless, meaning that concatenating all components in their original order reconstructs the original prompt exactly. Placeholders are extracted separately if they are knowledge input, user questions, or contextual information. Then, annotators determine if each of the segment is an instruction or few-shot examples. If neither applied, annotators will write a description of the segment and then select the most appropriate label from the remaining labels.

We denote the subset of synthetic prompts as `Synthetic` and the subset of real-world prompts as `Langchain`. Tab. 1 reports the statistics of these subsets, and Fig. 8 shows the distribution of segment labels. For the `Synthetic` subset, we use 14,732 prompts for training, 200 for validation, and 200 for testing. For the `Langchain` subset, all prompts are used exclusively for testing.

| | Prompts | Tokens | Segments |
|---|---|---|---|
| Synthetic | 15,132 | 900 | 6.1 |
| Langchain | 197 | 914 | 7.6 |

Table 1: Dataset statistics comparing synthetic and real-world prompts.

## 5.2 IMPLEMENTATION DETAILS

We evaluate `BoundRL` on three LLMs: Qwen3-1.7b, Qwen3-4b (Yang et al., 2025), and Llama-3.1-8b-Instruct (Dubey et al., 2024). The training process has two stages:

**Stage 1: Supervised Fine-tuning (SFT)** All LLMs are first fine-tuned on the training set of `StructSeg` for one epoch with a batch size of 16. We use learning rates of 2e-6 for Qwen3 models and 5e-7 for Llama-3.1-8b-Instruct.

**Stage 2: Reinforcement Learning with Verifiable Rewards (RLVR)** SFT-tuned models are then tuned with RLVR using GRPO (Shao et al., 2024) without standard deviation-based reward scaling (Liu et al., 2025). To control computational costs, we use a randomly sampled 25% subset of the training data. Each training batch contains 6 input documents, with $m = 4$ candidate segmentation generated per input text using temperature of 1.2 during rollout. For intermediate candidate con-

---

[2]https://smith.langchain.com/hub

| Method | Synthetic | | | | | Langchain | | | | | Avg |
|---|---|---|---|---|---|---|---|---|---|---|---|
| | $\rho_{\text{rec}}$ | EM | $P_k$ | F1$_{\text{lab}}$ | F1$_{\text{char}}$ | $\rho_{\text{rec}}$ | EM | $P_k$ | F1$_{\text{lab}}$ | F1$_{\text{char}}$ | |
| Qwen3-1.7b | | | | | | | | | | | |
| SFT w/start | 99.3 | 72.3 | 4.6 | 94.1 | 93.7 | 87.7 | 34.7 | 14.7 | 77.0 | 71.1 | **81.1** |
| SFT w/end | 96.0 | 64.8 | 6.0 | 92.0 | 89.3 | 77.7 | 26.5 | 19.5 | 71.1 | 63.9 | 75.6 |
| SFT w/start+end | 96.6 | 59.5 | 7.9 | 90.8 | 86.7 | 84.5 | 20.2 | 17.9 | 69.9 | 65.2 | 74.8 |
| Qwen3-4b | | | | | | | | | | | |
| SFT w/start | 99.7 | 71.6 | 5.3 | 94.9 | 92.8 | 93.1 | 41.6 | 12.3 | 80.2 | 78.8 | **83.5** |
| SFT w/end | 98.1 | 67.8 | 4.6 | 93.7 | 92.5 | 91.5 | 39.9 | 13.0 | 79.0 | 78.5 | 82.3 |
| SFT w/start+end | 98.8 | 70.4 | 5.9 | 93.9 | 92.7 | 85.2 | 33.7 | 14.1 | 76.5 | 71.8 | 80.3 |
| Llama-3.1-8b-Instruct | | | | | | | | | | | |
| SFT w/start | 99.6 | 71.8 | 4.9 | 94.3 | 93.4 | 95.9 | 28.4 | 13.4 | 79.7 | 80.7 | **82.5** |
| SFT w/end | 98.7 | 65.9 | 5.6 | 93.8 | 92.5 | 93.5 | 31.5 | 13.6 | 75.6 | 76.6 | 80.9 |
| SFT w/start+end | 97.8 | 62.9 | 6.2 | 90.8 | 91.4 | 87.7 | 25.1 | 17.1 | 71.4 | 72.3 | 77.6 |

Table 2: Evaluation of different output patterns. The best-performing output pattern is highlighted in **bold**. SFT w/start, the output pattern used by `BoundRL`, consistently outperforms other patterns.

struction, we apply selective replacement with model-specific thresholds: $k = 2$ for Qwen3 models and $k = 1$ for Llama-3.1. Learning rates are 1e-6 for Qwen3 models and 2e-7 for Llama-3.1. We checkpoint every 0.2 epochs and select the best model based on validation performance.

During inference, the temperature is set to 0. We tune the hyperparameters based on their performance on the validation set. More implementation details are in App. A.3.

# 6 EXPERIMENT RESULTS

## 6.1 EVALUATION OF OUTPUT PATTERNS

In this section, we evaluate output patterns for structured text segmentation by comparing LLMs fine-tuned with SFT for different output patterns. We consider three output patterns: (i) SFT w/start, which is used by `BoundRL` and outputs starting tokens of each segment; (ii) SFT w/end, which outputs ending tokens of each segment; (iii) SFT w/start+end, which outputs both starting and ending tokens of each segment. We show examples of these output patterns in App. A.5.

For evaluation, we use the reconstruction ratio $\rho_{rec}(*)$, the F-1 score of exact match EM($*$) and the character-level F-1 score F1$_{\text{char}}(*)$ as described in Sec. 4.2. We additionally use character-level $P_k$ score (Beeferman et al., 1999) which measure the quality of segment boundaries, with the window width set to half the average length of annotated segments following standard practice. We also use $F1_{lab}$, which is the micro-F1 score that compares the predicted label of each generated segment with the label of the most overlapping annotated segment following Arnold et al. (2019a). Higher values of all metrics except $P_k$ indicate better performance, while lower $P_k$ values are better. Results are reported in percentage on the test set of the Synthetic subset and the Langchain subset, along with the average of one minus $P_k$ and other metrics across both subsets in Tab. 2.

Table 2 shows that SFT w/start, used by `BoundRL`, consistently outperforms other output patterns across LLMs and datasets, with particular advantages in exact match scores. Contrarily, SFT w/start+end performs worse than both SFT w/start and SFT w/end, although it is supposed to be more robust to token mismatches as it generates both boundaries of each segment. This suggests that requiring simultaneous generation of starting and ending tokens imposes an excessive learning burden that degrades performance.

## 6.2 EVALUATION OF BOUNDRL

In this section, we perform a comprehensive evaluation of `BoundRL`. We consider the following training schemes: (i) SFT, where models are fine-tuned with SFT for one epoch to adapt the output pattern of `BoundRL`; (ii) SFT w/2 epochs, where models are fine-tuned with SFT for two epochs; (iii) NER, where models are fine-tuned for two epochs to predict the label for each token in the prompt like named entity recognition (NER) (Li et al., 2020); (iv) SFT+RLVR, a two-stage fine-tuning procedure as in `BoundRL`, but without intermediate candidates; (v) SFT+RLVR$_{\text{w/ high temp.}}$, the same as SFT+RLVR but with a higher sampling temperature of 1.5 during rollout; (vi) RL-

| Method | Synthetic | | | | | Langchain | | | | | Avg |
|---|---|---|---|---|---|---|---|---|---|---|---|
| | $\rho_{rec}$ | EM | $P_k$ | $F1_{lab}$ | $F1_{char}$ | $\rho_{rec}$ | EM | $P_k$ | $F1_{lab}$ | $F1_{char}$ | |
| Prompting Baselines | | | | | | | | | | | |
| Claude3.5-Sonnet$_{full}$ | 78.3 | 14.3 | 15.4 | 79.8 | 70.2 | 55.8 | 11.0 | 21.4 | 62.4 | 47.5 | 58.2 |
| Claude3.5-Sonnet$_{start}$ | 50.0 | 16.9 | 25.2 | 73.8 | 47.2 | 48.1 | 11.4 | 23.4 | 61.3 | 41.1 | 50.1 |
| Claude4-Sonnet$_{full}$ | 97.2 | 22.1 | 11.3 | 82.2 | 88.2 | 80.3 | 13.8 | 18.1 | 65.5 | 68.3 | 68.8 |
| Calude4-Sonnet$_{start}$ | 90.1 | 22.8 | 13.6 | 79.8 | 81.8 | 87.0 | 18.3 | 18.1 | 67.9 | 71.6 | 68.8 |
| Qwen3-1.7b | | | | | | | | | | | |
| SFT | 99.3 | 72.3 | 4.6 | 94.1 | 93.7 | 87.7 | 34.7 | 14.7 | 77.0 | 71.1 | 81.1 |
| SFT w/2epochs | 99.5 | 73.5 | 3.9 | 94.4 | 94.6 | 85.0 | 41.0 | 14.6 | 77.6 | 70.9 | 81.8 |
| NER | 100.0 | 34.1 | 6.5 | 81.5 | 94.8 | 100.0 | 8.9 | 19.9 | 57.4 | 86.7 | 73.7 |
| SFT+RLVR | 100.0 | 77.4 | 4.1 | 94.7 | 94.6 | 88.6 | 47.2 | 13.2 | 79.1 | 74.6 | 83.9 |
| SFT+RLVR$_{w/temp.}$ | 100.0 | 77.2 | 4.0 | 94.6 | 95.0 | 90.4 | 44.6 | 14.4 | 79.1 | 75.4 | 83.8 |
| RL-PLUS | 99.8 | 73.9 | 4.5 | 94.4 | 94.3 | 91.5 | 42.9 | 13.4 | 79.5 | 76.5 | 83.5 |
| BoundRL | 99.9 | 77.3 | 4.1 | 94.8 | 94.8 | 90.6 | 47.3 | 12.2 | 79.8 | 76.8 | **84.5** |
| Qwen3-4b | | | | | | | | | | | |
| SFT | 99.7 | 71.6 | 5.3 | 94.9 | 92.8 | 93.1 | 41.6 | 12.3 | 80.2 | 78.8 | 83.5 |
| SFT w/2epochs | 99.7 | 73.0 | 4.3 | 95.2 | 94.2 | 91.3 | 40.7 | 12.1 | 83.6 | 78.2 | 84.0 |
| NER | 100.0 | 41.9 | 6.9 | 82.8 | 95.6 | 100.0 | 8.9 | 24.7 | 59.3 | 85.7 | 74.3 |
| SFT+RLVR | 99.7 | 77.6 | 4.6 | 94.6 | 93.7 | 92.7 | 52.4 | 10.6 | 82.3 | 82.1 | 86.0 |
| SFT+RLVR$_{w/temp.}$ | 99.7 | 77.3 | 4.9 | 94.4 | 93.3 | 87.6 | 47.0 | 12.5 | 77.6 | 74.2 | 83.4 |
| RL-PLUS | 99.7 | 76.6 | 4.2 | 94.3 | 94.1 | 94.8 | 51.0 | 10.8 | 81.5 | 83.1 | 86.0 |
| BoundRL | 99.7 | 78.3 | 4.0 | 94.8 | 94.7 | 94.1 | 52.4 | 10.3 | 82.5 | 83.3 | **86.6** |
| Llama-3.1-8b-Instruct | | | | | | | | | | | |
| SFT | 99.6 | 71.8 | 4.9 | 94.3 | 93.4 | 95.9 | 28.4 | 13.4 | 79.7 | 80.7 | 82.5 |
| SFT w/2epochs | 99.9 | 72.8 | 4.5 | 94.3 | 94.2 | 95.6 | 31.9 | 13.2 | 78.9 | 79.5 | 82.9 |
| NER | 100.0 | 25.9 | 12.3 | 69.9 | 92.4 | 100.0 | 7.0 | 24.9 | 55.7 | 83.4 | 69.7 |
| SFT+RLVR | 100.0 | 73.9 | 4.1 | 94.7 | 94.6 | 96.4 | 40.2 | 11.7 | 77.3 | 82.1 | 84.3 |
| SFT+RLVR$_{w/temp.}$ | 99.7 | 72.7 | 4.1 | 94.0 | 94.6 | 91.8 | 43.3 | 13.0 | 77.5 | 78.9 | 83.5 |
| RL-PLUS | 100.0 | 73.0 | 4.4 | 94.4 | 94.3 | 95.9 | 37.9 | 11.7 | 78.0 | 82.7 | 84.0 |
| BoundRL | 100.0 | 76.1 | 4.4 | 94.4 | 94.1 | 96.3 | 42.8 | 11.5 | 78.0 | 82.1 | **84.8** |

Table 3: Evaluation of BoundRL across LLMs and datasets. The best-performing method for each LLM is highlighted in **bold**. BoundRL consistently outperform both finetuning baselines and few-shot prompting with much larger LLMs. The improvements are particularly big on the Langchain subset, showing BoundRL's superior generalization to real-world, out-of-domain prompts.

PLUS (Dong et al., 2025), which uses one sequence of annotated segments and three candidate segmentations during rollout. Implementation details of these baselines are in A.4. For comparison, we also consider few-shot prompting baselines using Claude3.5v2-sonnet (Anthropic, 2024) and Claude4-sonnet (Anthropic, 2025). In this setting, the LLM is instructed to segment an input prompt according to the target taxonomy and a provided example. We consider two output patterns: (i) full, which outputs the complete text of each segment; and (ii) start, which outputs only the starting tokens, as in BoundRL. The prompts used for these baselines are shown in Appendix A.2. The results are in Table 3. Qualitative examples of these baselines are in App. A.8.

We observe that BoundRL with intermediate candidate construction consistently outperforms all baselines. The difference between BoundRL and the second best-performing method (SFT+RLVR) is statistically significant using paired t-test ($p < 0.05$), showing the importance of intermediate candidates for effective RLVR training. We additionally show standard deviation of reward during training and find that BoundRLcan mitigate the entropy collapse issue of RLVR in App. A.7. In contrast, RL-PLUS, which uses annotated segments during rollout, has inconsistent results and can even hurt performance. This may be because annotated segments are too out-of-distribution to provide useful learning signals. Additionally, increasing temperature (SFT+RLVR$_{w/temp.}$) cannot further improve the performance, showing that the improvement brought by intermediate candidates is not merely from increased exploration space but guided exploration.

We also observe that LLMs fine-tuned with RLVR consistently outperforms SFT-only models. The difference between SFT+RLVR and SFT w/2epochs is a statistically significant using paired t-test ($p < 0.05$). Most notably, the improvement becomes bigger on the Langchain subset (real-world prompts), with RLVR showing 5-11% absolute improvements in exact match scores. Conversely, doubling SFT training epochs (SFT w/2 epochs) yields marginal improvements on the in-domain Synthetic subset but degrades performance on the Langchain subset for some LLMs, indicating overfitting. These results highlight RLVR's superior generalization to out-of-distribution data.

| Method | Synthesis | | | | | Langchain | | | | | Avg |
|--------|-----------|---|---|---|---|-----------|---|---|---|---|-----|
| | $\rho_{rec}$ | EM | $P_k$ | $F1_{lab}$ | $F1_{char}$ | $\rho_{rec}$ | EM | $P_k$ | $F1_{lab}$ | $F1_{char}$ | |
| Qwen3-1.7b | | | | | | | | | | | |
| BoundRL | 99.9 | 77.3 | 4.1 | 94.8 | 94.8 | 90.6 | 47.3 | 12.2 | 79.8 | 76.8 | **84.5** |
| BoundRL w/ 2steps | 99.9 | 78.0 | 4.4 | 94.8 | 94.7 | 88.6 | 45.7 | 13.6 | 80.5 | 75.0 | 83.9 |
| BoundRL w/o select | 99.7 | 76.9 | 4.5 | 94.6 | 94.1 | 89.9 | 45.3 | 12.2 | 79.8 | 76.5 | 84.0 |
| BoundRL w/o middle | 99.6 | 77.7 | 4.5 | 94.9 | 94.2 | 89.7 | 44.5 | 12.9 | 78.8 | 75.5 | 83.7 |
| Qwen3-4b | | | | | | | | | | | |
| BoundRL | 99.7 | 78.3 | 4.0 | 94.8 | 94.7 | 94.1 | 52.4 | 10.3 | 82.5 | 83.3 | **86.6** |
| BoundRL w/ 2steps | 99.7 | 78.3 | 3.7 | 94.5 | 94.7 | 94.7 | 50.0 | 10.9 | 81.5 | 84.8 | 86.4 |
| BoundRL w/o select | 99.7 | 77.4 | 4.5 | 94.8 | 93.6 | 93.7 | 52.1 | 10.5 | 83.0 | 84.0 | 86.3 |
| BoundRL w/o middle | 99.7 | 77.7 | 4.2 | 94.8 | 94.2 | 94.6 | 50.7 | 11.0 | 81.4 | 84.1 | 86.2 |
| Llama-3.1-8b-Instruct | | | | | | | | | | | |
| BoundRL | 100.0 | 76.1 | 4.4 | 94.4 | 94.1 | 96.3 | 42.8 | 11.5 | 78.0 | 82.1 | **84.8** |
| BoundRL w/2steps | 100.0 | 76.6 | 4.3 | 94.4 | 94.6 | 94.4 | 38.4 | 12.2 | 77.9 | 81.5 | 84.1 |
| BoundRL w/o select | 99.9 | 75.7 | 4.3 | 94.3 | 94.6 | 94.7 | 41.8 | 11.6 | 78.7 | 81.5 | 84.5 |
| BoundRL w/o middle | 99.9 | 74.7 | 4.4 | 94.0 | 94.2 | 95.5 | 43.2 | 12.1 | 77.6 | 81.8 | 84.4 |

Table 4: Ablation study of BoundRL. The best-performing method of each LLM is in **bold**.

We note that the smallest model (Qwen3-1.7b) fine-tuned with BoundRL significantly outperforms the best-performing few-shot prompting baseline (Claude4-sonnet-full) with much more parameters. The efficiency gains are also substantial. Prompting baselines that generate full segment text requires an average of 1,170 tokens per input prompt on the Synthetic subset, while BoundRL requires only 119 tokens, which corresponds to a 90% reduction in output tokens.

Although models fine-tuned with NER achieve high scores on the character-level F1, $F1_{char}$, they generally achieve low scores on exact match EM, $P_k$ and $F1_{lab}$. Analysis of the outputs shows that models fine-tuned with NER tend to generate very fragmented and short segments, demonstrating the effectiveness of framing structured text segmentation as a boundary generation task.

### 6.3 ABLATION STUDY OF BOUNDRL

In this section, we perform ablation study of BoundRL. We consider the following ablated versions of BoundRL: (i) BoundRL w/ 2steps, which performs two perturbation steps to candidate segmentations to generate intermediate candidates; (ii) BoundRL w/o select, which incorporates a intermediate candidate for each input text in a batch without selective replacement; (iii) BoundRL w/o middle, which generates intermediate candidates by perturbing a randomly sampled candidate segmentation instead of the one with the medium-level reward. Implementation details of these ablated versions are in App. A.9. The results are in Tab. 4.

Tab. 4 shows that BoundRL outperforms all ablated versions. Specifically, applying multiple perturbations when generating intermediate candidates (BoundRL w/ 2steps) and incorporating them for all input texts (BoundRL w/o select) both hurt performance. The results show the importance of controlling the distance between the current generation and intermediate candidates, which aligns with our findings in Sec. 6.2 that directly using annotated segments does not improve performance.

## 7 CONCLUSIONS

We propose BoundRL, a novel framework that reformulates structured text segmentation as a boundary generation problem. Instead of regenerating entire text segments, models generate only the starting tokens of each segment, which substantially reduces inference costs and mitigates hallucination risks. To adapt the model to this output format, BoundRL employs RLVR to jointly optimize reconstruction fidelity and semantic alignment, while our intermediate candidate construction strategy alleviates entropy collapse during training. In a challenging case study on LLM prompts, BoundRL consistently outperforms fine-tuning baselines using SFT and RLVR as well as the few-shot prompting baseline with much larger models.

Future work could extend the boundary generation paradigm to hierarchical document structures, such as legal texts and technical reports, and explore few-shot annotation methods to lower annotation effort in new domains. Due to its domain-agnostic design, BoundRL offers a foundation for efficient structured text analysis across the expanding landscape of complex document processing.

## 8 ETHICS STATEMENT

The human annotations were collected through hired annotators from a data annotation service. Annotators were instructed to strictly refrain from including any biased, hateful, or offensive content towards any race, gender, sex, or religion. The annotations passed through audits, where they were examined by a separate group of annotators and reached a 89% agreement ratio.

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

# A   APPENDIX

## A.1   TAXONOMY USED FOR PROMPTS

| Label | Definition |
|---|---|
| Instruction | Guidance on how to process and respond to queries. |
| Example | Examples of what the input and corresponding output should look like. |
| Context | Background information and context that the model needs to refer to. |
| Question | Queries or questions provided specifically by users |
| Output Format | The type, format, or style of the output |

Table 5: Example taxonomy for structured prompt segmentation used in `StructSeg`. Different domains may employ alternative taxonomies appropriate to their document types and analysis needs.

## A.2   IMPLEMENTATION DETAILS OF CLAUDE

In this section, we describe the implementation details for Claude3.5-sonnet-v2 and Claude4-sonnet for the prompt segmentation task. The prompts used by the models first give a detailed definition for each label used by our taxonomy and then instructs the model to extract segments following the definitions. As described in Sec. 6.2, the models are required to output either the full text of each segment or the starting tokens of each segment as `BoundRL`. To help model better understand the required output format, the prompts also include a randomly sampled prompt and its expected output format from the training set of `StructSeg`. The prompt for outputting the full text of each segment is shown in Figs 4. The prompt for outputting the starting tokens of each segment is shown in Fig. 4. The temperature during inference is set to 0.

## A.3   IMPLEMENTATION DETAILS OF BOUNDRL

In this section, we describe additional implementation details of `BoundRL`. To help model better adapt to the prompt segmentation task, we give models a meta instruction in addition to the prompt to be segmented. The meta instruction includes a brief definition of each segment type and an example of required output format. The meta instruction that requires the model to output starting tokens of each segment is shown in Fig. 5.

For both SFT and reinforcement learning, `BoundRL` sets the maximum gradient norm as 0.1 and the weight decay as 0.01. `BoundRL` uses the linear learning rate scheduler with a warmup ratio of 0.03. We tune the hyperparameters of `BoundRL` in stages. We first select the hyperparameters of SFT based on the performance of models that are finetuned SFT on the validation set. With the SFT hyperparameters fixed, we then tune the hyperparameters of RLVR, followed by those for intermediate candidate construction, using the same process.

During the rollout stage of reinforcement learning, we notice that candidate segmentation might contain repetitive end of response tokens or segments after end of response tokens, which might hurt the training stability if are directly used for training. To address the issue, all generated candidate segmentations are truncated at the first end of response token.

To shorten or extend the text of a segment $\hat{t}_i$, `BoundRL` modifies the starting token sequences $\hat{s}_i$ or $\hat{s}_{i+1}$ accordingly. Specifically, to shorten the text of a segment $\hat{t}_i$ on the left side by one word, `BoundRL` truncates the first word of the starting token sequence $\hat{s}_i$. To shorten the text of a segment $\hat{t}_i$ on the right side by one word, `BoundRL` prepends to $\hat{s}{i+1}$ the word immediately before it. To extend the text of a segment $\hat{t}_i$ on the right side by one word, `BoundRL` truncates the first word of the starting token sequence $\hat{s}_{i+1}$. To extend the text of a segment $\hat{t}_i$ on the left side by one word, `BoundRL` prepends to $\hat{s}i$ the word immediately before it. Therefore, When constructing intermediate candidates, `BoundRL`does not shorten or extending the text of a segment with only one word. We will also modify the neighboring starting token sequences accordingly if there is a overlap between starting token sequences after modifications.

### A.4 IMPLEMENTATION DETAILS OF BASELINES

Unless otherwise specified, SFT and SFT w/2epochs use the same hyper-parameters as the SFT training stage of `BoundRL`. The label schema of our NER baseline is motivated by (Tjong Kim Sang & De Meulder, 2003), which uses 'B-X', 'I-X', and 'O'. However, since all tokens in a prompt belong to a segment in the text segmentation task, we use 'B-X' to represent the beginning token of each segment and 'I-X' to represent the remaining tokens of each segment. To predict the label of each token, the output of the final layer for each token is feed into a single-layer MLP following the common practice. Other hyper-parameters of the NER baseline is the same as the SFT training stage of `BoundRL`. For RL-PLUS, we replace one originally generated candidate segmentation with a sequence of annotated segments. The temperature to control the weight of advantage function is 1.0. Unless otherwise specified, RL-PLUS, SFT+RLVR, and SFT+RLVR$_{\text{w/ high temp.}}$ all use GRPO without reward scaling based on standard deviation and the same hyper-parameters as `BoundRL` for a fair comparison.

### A.5 IMPLEMENTATION DETAILS OF OTHER OUTPUT PATTERNS

In this section, we provide implementation details for output patterns other than the output pattern used by `BoundRL` ('start'). Specifically, for the 'end' output pattern, the model should first generate a label and then a sequence of ending tokens for each segment. The text of the $i$-th segment $\hat{t}_i$ is then extracted as the text span between the positions of the $i-1$-th and $i$-th sequence of ending tokens. For the 'start+end', the model should first output a label and then output a sequence of starting tokens and a sequence of ending tokens. The text of the $i$-th segment $\hat{t}_i$ is then extracted as the text span between the positions of the $i$-th sequence of starting tokens and the $i$-th sequence of ending tokens. We show the meta instruction that requires the model to output ending tokens of each segment in Fig. 6 and meta instruction that requires the model to output both starting and ending tokens of each segment in Fig. 7. We also show an example text and corresponding expected outputs for different output patterns in Fig. 9.

### A.6 GENERATION OF SYNTHETIC PROMPTS

When generating the synthetic prompts, we implement a multi-faceted sampling strategy that draws from varied prompt types (system prompt, user prompt, combined), prompt modes (prompt template, full prompt, hybrid), and task types (e.g. classification, summarization), placeholder formats (e.g., {context} or {{context}}), the number of examples (zero, one, few ), prompt lengths, writing styles, and levels of details. The full list of the task types, writing styles, prompt lengths, format types, and level of details are shown in Tab. 6.

### A.7 STANDARD DEVIATION OF REWARDS DURING TRAINING

In this section, we compare the standard deviation of rewards of SFT+RLVR and `BoundRL` during training to evaluate whether `BoundRL` can mitigate the entropy collapse issue of RLVR. Specifically, we show the curve of standard deviation of rewards among candidate segmentations generated during rollout along the training. We show the curve of SFT+RLVR and `BoundRL` in Fig.10. From the figure, we find that the standard deviation of rewards of SFT+RLVR quickly becomes very small, while that of `BoundRL` remains stable throughout training. The results show that intermediate candidates help `BoundRL` mitigate the entropy collapse issue of RLVR.

### A.8 QUALITATIVE EXAMPLES

In this section, we show qualitative examples of `BoundRL` and other baselines. Specifically, we an example prompt, its corresponding annotated segments, the raw output and reconstructed segments for each method in Fig. 11.

### A.9 IMPLEMENTATION DETAILS OF ABLATION STUDIES

In this section, we provide more implementation details of the ablated versions of `BoundRL`. In `BoundRL` w/ 2steps, we construct the intermediate candidates by selecting the first perturbation

step that has the biggest reward gain over the original candidate segmentation with the medium-level reward. We then select the second perturbation that has the biggest reward gain when the first perturbation step is applied. The intermediate candidate is then constructed by applying the first and second perturbation steps on the original candidate segmentation. For `BoundRL` w/o select, we incorporate intermediate candidates for all input texts where the intermediate candidate has a positive reward gain over the original generated candidate segmentation, rather than restricting replacement to the top-$k$ cases. For `BoundRL` w/o middle, we construct intermediate candidates by perturbing a randomly sampled candidate segmentation instead of the one with the medium-level reward. Other design choices for these ablated versions are the same as those for `BoundRL` for a fair comparison.

### A.10 USAGE OF LLMS IN WRITING

In this paper, we use LLMs solely to polish our draft. We do not use LLMs for research ideation, retrieval, or discovery.

Segment the following prompt into different categories and referring to their descriptions:
<categories>
* <option>instruction</option>: Including 1/ profile/role that the model is acting as; 2/ Core intent of the prompt; 3/ workflow or steps and processes the model should follow to complete the task; 4/ restrictions on what the model must adhere to when generating responses
* <option>context</option>: Background information and context that the model needs to refer to. This can include 1/ background or supplementary input that helps set the stage for the task but is not the primary focus. 2/ Knowledge input - The core content that the prompt directly processes or manipulates; 3/ Metadata/Short Phrases - Brief inputs or settings that define specific parameters or goals for the task.
* <option>question</option>: Queries or questions provided specifically by users. The questions that are part of the template should not be labeled as question but as instruction.
* <option>examples</option>: Providing the AI model with concrete examples of the desired input-output pattern before asking it to perform a similar task. These examples demonstrate the expected format, style, and reasoning pattern, helping the model understand and replicate the desired behavior. A example must contain both concrete input and output. If either input or output is missing, it should not be labeled as example but be labeled as . If input or output does not have actual content, it should not be labeled as example.
* <option>output_format</option>: Specific requirement of the type, format, or style of the output, such as the exact json format or function calling language. A general requirement like 'the output should be in json format' should not be labeled as output_format.
</categories>
<notes>
* Do not separate consecutive prompt segments if they belong to the same category. Make them into one component.
* If unclear, or you are looking at less meaningful text pieces between different components, label it as instruction
* Must keep the text of each component exactly the same as the original prompt.
* Your response between <segmentation_annotation> </segmentation_annotation> must be parsable by Python's ast.literal_eval(). Avoid use single quotes within text of each component.
</notes>
<example>
<example_prompt>
{example_prompt}
</example_prompt>
<example_segmentation>
{example_segmentation}
</example_segmentation>
</example>
<prompt_to_analyze>
{prompt_to_analyze}
</prompt_to_analyze>
<segmentation_annotation>
[{{
  'relative_order': 0,
  'text': [text of component 1],
  'type': [type of component 1, given categories only],
}},
...
{{
  'relative_order': N,
  'text': [text of component N],
  'type': [type of component N, given categories only],
}},
]
</segmentation_annotation>
provide your annotated answer enclosed in <segmentation_annotation></segmentation_annotation> XML tags.

Figure 3: Prompt used by Claude to extract the full text of each segment

Segment the following prompt into different categories and referring to their descriptions:

<categories>

* <option>instruction</option>: Including 1/ profile/role that the model is acting as; 2/ Core intent of the prompt; 3/ workflow or steps and processes the model should follow to complete the task; 4/ restrictions on what the model must adhere to when generating responses

* <option>context</option>: Background information and context that the model needs to refer to. This can include 1/ background or supplementary input that helps set the stage for the task but is not the primary focus. 2/ Knowledge input - The core content that the prompt directly processes or manipulates; 3/ Metadata/Short Phrases - Brief inputs or settings that define specific parameters or goals for the task.

* <option>question</option>: Queries or questions provided specifically by users. The questions that are part of the template should not be labeled as question but as instruction.

* <option>examples</option>: Providing the AI model with concrete examples of the desired input-output pattern before asking it to perform a similar task. These examples demonstrate the expected format, style, and reasoning pattern, helping the model understand and replicate the desired behavior. A example must contain both concrete input and output. If either input or output is missing, it should not be labeled as example but be labeled as . If input or output does not have actual content, it should not be labeled as example.

* <option>output_format</option>: Specific requirement of the type, format, or style of the output, such as the exact json format or function calling language. A general requirement like 'the output should be in json format' should not be labeled as output_format.

</categories>

<notes>

* Do not separate consecutive prompt segments if they belong to the same category. Make them into one segment.

* If unclear, or you are looking at less meaningful text pieces between different segments, label it as instruction

* Only output starting words (less than 10 words) of each segment instead of outputting the whole segment

* Must keep the starting words of each segment exactly the same as the corresponding words of the original prompt.

* The extracted segments should be in the order as they are in the original prompt.

* Your response between <segmentation_annotation> </segmentation_annotation> must be parsable by Python's ast.literal_eval(). Avoid use single quotes within text of each component.

</notes>

<example>

<example_prompt>

{example_prompt}

</example_prompt>

<example_segmentation>

{example_segmentation}

</example_segmentation>

</example>

<prompt_to_analyze>

{prompt_to_analyze}

</prompt_to_analyze>

<segmentation_annotation>

```
[{{
  'relative_order': 0,
  'text': [starting words of segment 1],
  'type': [type of segment 1, given categories only],
 }},
 ...
 {{
  'relative_order': N,
  'text': [starting words of segment N],
  'type': [type of segment N, given categories only],
 }}]
```

</segmentation_annotation>

provide your annotated answer enclosed in <segmentation_annotation></segmentation_annotation> XML tags.

Figure 4: Prompt used by Claude to extract the starting tokens of each segment

You are requested to segment a prompt into following categories:

(1) instruction: A guidance on how to process and respond to queries.

(2) context: Background information and context that the model needs to refer to.

(3) question: Queries or questions provided specifically by users.

(4) examples: Examples of what the input / output should look like.

(5) output_format: The type, format, or style of the output.

You should output the segments of a prompt in the form of [category_1]start_tokens_1%<separator>%\n[category_2]start_tokens_2...\n[category_N]start_tokens_N, where 'category_1' denotes the predicted category for segment 1 and 'start_tokens_1' denotes the start tokens of segment 1.

Given the following LLM prompt, please follow the instruction above to complete the data field extraction task.

Figure 5: Meta instruction used by `BoundRL` to output starting tokens of each segment.

You are requested to segment a prompt into following categories:

(1) instruction: A guidance on how to process and respond to queries.

(2) context: Background information and context that the model needs to refer to.

(3) question: Queries or questions provided specifically by users.

(4) examples: Examples of what the input / output should look like.

(5) output_format: The type, format, or style of the output.

You should output the segments of a prompt in the form of [category_1]end_tokens_1%<separator>%\n[category_2]end_tokens_2...\n[category_N]end_tokens_N, where 'category_1' denotes the predicted category for segment 1 and 'end_tokens_1' denotes the end tokens of segment 1.

Given the following LLM prompt, please follow the instruction above to complete the data field extraction task.

Figure 6: Meta instruction used by `BoundRL` to output ending tokens of each segment.

You are requested to segment a prompt into following categories:

(1) instruction: A guidance on how to process and respond to queries.

(2) context: Background information and context that the model needs to refer to.

(3) question: Queries or questions provided specifically by users.

(4) examples: Examples of what the input / output should look like.

(5) output_format: The type, format, or style of the output.

You should output the segments of a prompt in the form of [category_1]start_tokens_1%<separator>%end_tokens_1[/category_1]\n...\n[category_N]start_tokens_N%<separator>%end_tokens_N[/category_N], where 'category_1' denotes the predicted category for segment 1, 'start_tokens_1' denotes the start tokens of segment 1, and 'end_tokens_1' denotes the end tokens of segment 1.

Given the following LLM prompt, please follow the instruction above to complete the data field extraction task.

Figure 7: Meta instruction used by `BoundRL` to output both starting and ending tokens of each segment.

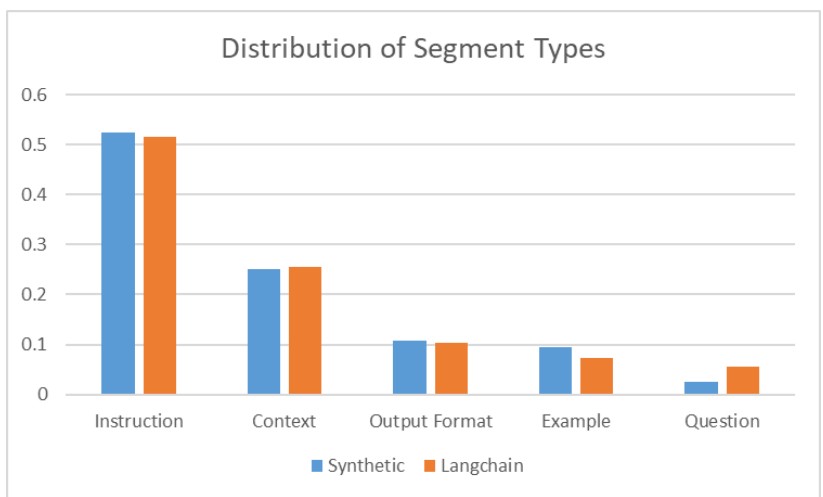

Figure 8: Distribution of segment labels across our dataset showing the proportion of each label type in both synthetic prompts (`Synthetic`) and real-world prompts (`Langchain`).

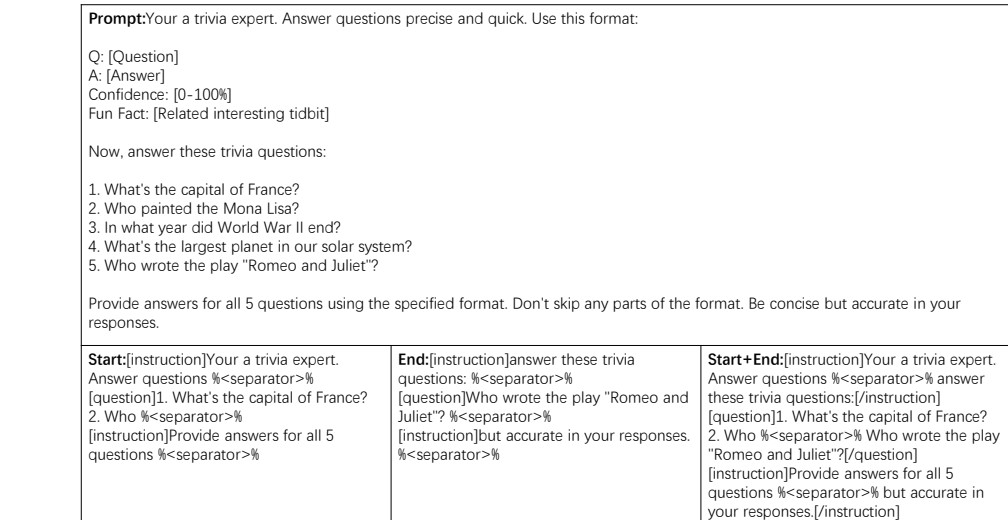

Figure 9: An example text and expected outputs for different output patterns.

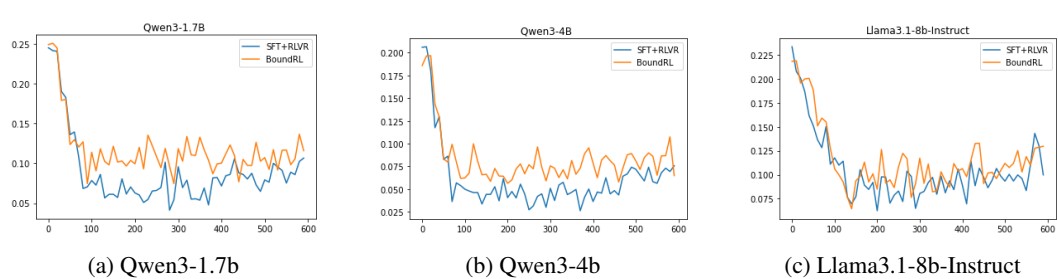

(a) Qwen3-1.7b  (b) Qwen3-4b  (c) Llama3.1-8b-Instruct

Figure 10: The standard deviation of rewards during training for `BoundRL` and SFT+RLVR. Intermediate candidates help `BoundRL` mitigate the entropy collapse issue of RLVR.

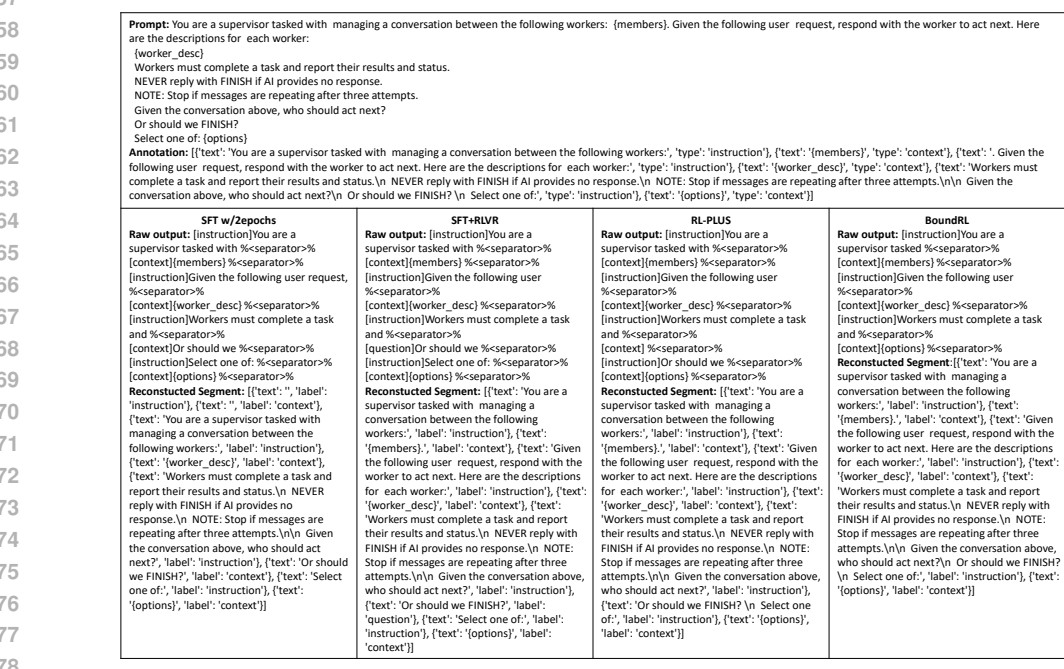

Figure 11: Qualitative examples of `BoundRL` and other baselines.

| Factor | Values |
|---|---|
| task type | 'named entity recognition', 'current events knowledge', 'document similarity comparison', 'bug detection', 'grammar and spell checking', 'code refactoring', 'parameter extraction', 'software development', 'webhook handling', 'clinical note summarization', 'table relationship inference', 'trivia answering', 'anything you can think that a user might need help with', 'web development', 'general productivity assistant', 'spam detection', 'scientific concept explanation', 'mathematical problem solving', 'email thread summarization', 'multi-option reasoning', 'genre classification', 'research paper abstracting', 'classification', 'database schema understanding', 'general coding assistance', 'text simplification', 'keyword extraction', 'news categorization', 'multi-API orchestration', 'fact-checking', 'sentiment analysis', 'citation finding', 'meeting minutes generation', 'news article summarization', 'fact verification', 'legal document summarization', 'complex query generation', 'toxicity detection', 'factoid QA', 'technical domain QA', 'semantic search', 'geographic knowledge QA', 'logical reasoning tasks', 'text generation', 'text style transfer', 'function calling', 'contextual recommendations', 'language detection', 'api authentication', 'evidence extraction', 'table-based QA', 'data filtering', 'code review', 'biographical information retrieval', 'open_book_qa (RAG), where a document is provided and a quesiton must be answered, but don\'t explicitly mention "open book qa"', 'dialogue summarization', 'topic classification', 'code explanation', 'code security enhancements', 'data type validation', 'anomaly detection in text', 'common sense reasoning', 'code generation', 'metaphor generation and interpretation', 'document type classification', 'ai coding assistant', 'poetry and song lyrics generation', 'customer feedback summarization', 'general programming ai assistance', 'code completion', 'chart/graph interpretation', 'automated essay scoring', 'paraphrasing', 'closed_book_qa, but don\'t explicitly mention "closed book qa"', 'multi-hop reasoning', 'input sanitzation', 'content moderation', 'SQL query optimization', 'api endpoint selection', 'context-dependent reasoning', 'error handling', 'intent classification', 'summarization', 'emotion classification', 'cross-document QA', 'multi-document summarization', 'text2sql', 'text completion', 'code summarization', 'reading comprehension', 'historical fact retrieval', 'video transcript summarization' |
| writing styles | 'to contain several noticeable grammatical errorsin direct and curt way', 'to contain several noticeable grammatical errors', 'in direct and curt way', 'to have lots of typos', 'in a well-formed style' |
| format type | 'a mixture of markdown and formatting seen in the example prompts provided above', 'a mixture of a formatting structure of your choice and section subtitles', 'a mixture of JSON or nested JSON and other', 'markdown', 'a mixture of JSON or nested JSON and markdown', 'YAML-style formatting', 'a mixture of a formatting structure of your choice and other', 'a mixture of markdown and section headers', 'a mixture of formatting seen in the example prompts provided above and section subtitles', 'a mixture of other and markdown', 'a mixture of XML tags and coding', 'a mixture of coding and XML tags', 'a mixture of section subtitles and a formatting structure of your choice', 'a mixture of JSON or nested JSON and coding', 'a mixture of a formatting structure of your choice and markdown', 'a mixture of section subtitles and coding', 'a mixture of formatting seen in the example prompts provided above and JSON or nested JSON', 'a mixture of XML tags and section subtitles', 'a mixture of XML tags and formatting seen in the example prompts provided above', 'a mixture of section headers and formatting seen in the example prompts provided above', 'a mixture of a formatting structure of your choice and coding', 'pseudo-code', 'a mixture of coding and a formatting structure of your choice', 'a mixture of coding and JSON or nested JSON', 'a mixture of markdown and coding', 'a mixture of other and JSON or nested JSON', 'a mixture of XML tags and a formatting structure of your choice', 'section subtitles', 'a mixture of other and a formatting structure of your choice', 'a mixture of section headers and section subtitles', 'a mixture of other and coding', 'a mixture of section subtitles and markdown', 'a mixture of section subtitles and other', 'a mixture of coding and section headers', 'a mixture of section headers and a formatting structure of your choice', 'a mixture of section headers and coding', 'chain-of-thought styling', 'capital letters to highlight important details', 'a mixture of other and section subtitles', 'a mixture of XML tags and JSON or nested JSON', 'XML tags', 'a mixture of coding and other', 'a mixture of other and formatting seen in the example prompts provided above', 'a mixture of formatting seen in the example prompts provided above and section headers', 'a mixture of a formatting structure of your choice and XML tags', 'a mixture of formatting seen in the example prompts provided above and other', 'a formatting structure of your choice', 'a mixture of section headers and XML tags', 'a mixture of XML tags and markdown', 'a mixture of JSON or nested JSON and a formatting structure of your choice', 'a mixture of markdown and a formatting structure of your choice', 'JSON or nested JSON', 'a mixture of a formatting structure of your choice and section headers', 'a mixture of JSON or nested JSON and section headers', 'a mixture of coding and formatting seen in the example prompts provided above', 'a mixture of JSON or nested JSON and formatting seen in the example prompts provided above', 'a mixture of section subtitles and section headers', 'a mixture of JSON or nested JSON and XML tags', 'a mixture of other and XML tags', 'a mixture of XML tags and other', 'a mixture of section headers and JSON or nested JSON', 'a mixture of markdown and JSON or nested JSON', 'a mixture of a formatting structure of your choice and formatting seen in the example prompts provided above', 'a mixture of section subtitles and XML tags', 'table-based formatting', 'a mixture of a formatting structure of your choice and JSON or nested JSON', 'a mixture of section headers and other', 'a mixture of formatting seen in the example prompts provided above and markdown', 'a mixture of formatting seen in the example prompts provided above and a formatting structure of your choice', 'a mixture of section headers and markdown', 'a mixture of markdown and section subtitles', 'a mixture of markdown and XML tags', 'tree-style hierarchical formatting', 'a mixture of section subtitles and formatting seen in the example prompts provided above', 'a mixture of coding and markdown', 'a mixture of section subtitles and JSON or nested JSON', 'coding', 'a mixture of other and section headers', 'a mixture of coding and section subtitles', 'a mixture of markdown and other', 'formatting seen in the example prompts provided above', 'a mixture of XML tags and section headers', 'a mixture of formatting seen in the example prompts provided above and XML tags', 'section headers', 'a mixture of JSON or nested JSON and section subtitles', 'a mixture of formatting seen in the example prompts provided above and coding' |
| prompt length | 'less than 150 words', '150 to 500 words', '500 to 1000 words', 'around 1000 words', '1000 to 2000 words' |
| level of detail | 'basic level of detail, meaning it can just give minimal descriptions of things', 'moderate level of detail, meaning it goes a bit in-depth into things', 'detailed, meaning you should describe things thoroughly and do not give short names or descriptions', 'very detailed, meaning everything is described very in-depth and production-level detail is included', 'extremely technically detailed, meaning as many specific details should be present as possible, including technical jargon, production-level of context, and complicated descriptions' |

Table 6: Full list of factors and corresponding potential values used for generating of synthetic prompts.

