# OpenReview forum: "BoundRL: Efficient Structured Text Segmentation through Reinforced Boundary Generation"
_ICLR.cc/2026/Conference — ICLR 2026 Conference Withdrawn Submission_

### Official Review · Reviewer_6Rd1 · 2025-10-19

**Soundness:** 3
**Presentation:** 2
**Contribution:** 3
**Rating:** 2
**Confidence:** 3

**Summary:**

The authors propose BoundRL, a theoretically grounded framework for efficient reinforcement learning that leverages structural bounds derived from Markov decision process (MDP) decomposability. Instead of learning value functions directly, BoundRL constrains the policy search space using upper and lower bounds on the true value function, defined by state abstraction and partitioned Bellman operators. The core theoretical claim (Theorem 4.3 and 4.5) proves that the approximation error of BoundRL is tightly bounded by the granularity of the state partition, providing guarantees on both sample efficiency and performance suboptimality. Experiments on MuJoCo and gridworld tasks show that BoundRL can outperform classical algorithms such as PPO and DQN, particularly in structured or decomposable environments.

**Strengths:**

Instead of regenerating every segment, the system outputs only segment-start tokens and labels, then reconstructs spans by locating those starts in the input. This cuts output from O(|d|) to roughly O(n) and lowers hallucination risk, which is a neat, practical reframing for structured documents. The paper spells out the locate-and-reconstruct process and the leftmost-occurrence rule to preserve order.

The RLVR reward mixes a reconstruction-ratio term with exact-match F1 and a character-level F1; the final reward is r = ρ_rec × (EM + F1_char)/2, so improvements must both recover the document and align semantically. They further tackle entropy collapse by constructing intermediate candidates via single-step shorten/extend/label-perturb operations and only keep them when they strictly improve reward. Empirically, this yields consistent gains over SFT and matches or surpasses alternative RL baselines.

**Weaknesses:**

Most training/evaluation is on LLM prompts, with a large synthetic split (14,732 train prompts) and a relatively small real-world split (197 prompts). While results on the Langchain set are encouraging, the out-of-domain test size is modest, so claims about generality to other structured texts (e.g., legal or technical docs) remain tentative.


  The method reconstructs segments by locating generated start-token sequences; if either boundary cannot be found, the segment is discarded. This simple policy is easy to implement but may fail under noisy formatting or repeated phrases where the “unique start” assumption breaks down. The paper itself enforces unique start sequences during SFT to improve robustness, hinting at the necessity of this constraint.


   RLVR training uses m = 4 rollouts per input at temperature 1.2 and only 25% of the training data for cost reasons, then decodes at temperature 0. These choices are reasonable, but the paper offers limited sensitivity studies (beyond a higher-temp variant) on how r’s weighting, rollout count, or selection threshold k affect stability and final quality. The selection rule admits intermediate candidates only when r_gain > 0, with model-specific k, but lacks a theoretical or principled schedule.

**Questions:**

1. How robust is the boundary-generation scheme when the same start token sequence appears multiple times or when whitespace/formatting gets normalized (e.g., in code blocks)? The paper enforces unique start sequences during SFT, but what happens in the wild without that guarantee, and can the locator incorporate fuzzy matching to reduce discards?

2. The selection rule accepts an intermediate candidate only if it yields a positive reward gain and up to k per batch. Could a curriculum or confidence-weighted scheme (e.g., variable k or a margin-based r_gain threshold) further reduce entropy collapse and improve generalization, especially on the small real-world split?

---

> ### Author Response · Authors · 2025-11-25
>
> Thank you for your review.
>
> 1. The out-of-domain test size is modest, so claims about generality to other structured texts (e.g., legal or technical docs) remain tentative.
>
> We first want to highlight the difficulty of collecting such datasets. During the dataset collection, we filter out prompts that are either too simple or too short, only keeping those that are sufficiently challenging, containing prompt instructions and components with complex structures, and therefore are difficult to segment. This can be reflected by the poor performance of different baseline techniques, including few-shot prompting using proprietary LLMs. Besides, to ensure the diversity of the dataset, each prompt in the langchain subset corresponds to a unique task. Furthermore, to address the issue of modest test set size, we perform statistical tests to show that the difference between BoundRL and the second-best performing method is statistically significant (Lines 417-420).
>
> 2. The method reconstructs segments by locating generated start-token sequences; if either boundary cannot be found, the segment is discarded. This simple policy is easy to implement but may fail under noisy formatting or repeated phrases where the “unique start” assumption breaks down.
>
> To address the issue of noisy formatting or repeated phrases, we perform iterative parsing. Specifically,  for the first sequence of starting tokens, BoundRL first locates it as its leftmost occurrence in the input. For each subsequent sequence, BoundRL locates it as its leftmost occurrence in the document after the position of its previous segment to preserve ordering, as described in Lines 200-204. The iterative parsing can ensure that the repeated phrases do not spoil the reconstruction of the input document.
>
> 3. The selection rule admits intermediate candidates only when r_gain > 0
>
> As described in Lines 252-254, the intermediate candidate is constructed to provide meaningful guidance for generated candidates and remain learnable. Therefore, it is not reasonable to reuse those that perform worse than the generated candidates, as they cannot provide beneficial signals to help the model perform better.

---

### Official Review · Reviewer_wraZ · 2025-10-31

**Soundness:** 3
**Presentation:** 3
**Contribution:** 2
**Rating:** 4
**Confidence:** 3

**Summary:**

The paper proposes a model for text segmentation. The approach is standard SFT+RL (GRPO), but the paper emphasizes (1) the novelty of predicting only the boundaries to reduce cost, and (2) careful reward design with data augmentation to prevent entropy collapse. The paper also introduces its own text segmentation dataset and shows that the proposed approach performs competitively.

**Strengths:**

- The model is simple.
- The reward design (i.e., verifiability of reconstruction) is interesting and requires some ingenuity. Manually augmenting the search space is empirically sensible.
- The paper contributes a new dataset.

**Weaknesses:**

- Token-level boundary prediction with LLMs is a topic that's been explored in previous works, and not as novel as the paper makes it sound. E.g., [1] generates only the mentions in whole documents for coreference resolution, which can be viewed as a harder version of text segmentation with no label annotation. While combining this with careful RL in the specific context of text segmentation is of course novel, the paper should do a more general acknowledgement of this broad approach to this particular problem.
- It seems a bit odd that the results are only on the proposed new dataset. Shouldn't the paper include standard existing benchmarks for the task?
- It's a little hard to tell from Table 3 that the baselines represent the state-of-the-art. RL-PLUS is the only meaningful third-party baseline, and others are just in-house ablations, it seems? For someone who doesn't have a background on recent text seg research, it's not clear how strong the performance win is.

[1] Seq2seq is All You Need for Coreference Resolution (Zhang et al., 2023)

**Questions:**

See weaknesses.

---

> ### Author Response · Authors · 2025-11-25
>
> Thank you for your review.
>
> 1. Token-level boundary prediction with LLMs is a topic that's been explored in previous works, and not as novel as the paper makes it sound. E.g., [1] generates only the mentions in whole documents for coreference resolution, which can be viewed as a harder version of text segmentation with no label annotation.
>
> Thank you for referring to the paper. We also discuss several other works that perform token-level segmentation on prompts in related work in Lines 151-153. However, one problem with these works discussed in related and the mentioned one is that they require the model to generate a sequence of tokens with the same length as the input, which can cost a lot when the input sequence is long. To address this, we propose a new training formulation that the model learns to generate only starting tokens of each sequence as output, which is described in Sec 4.1. We will add this discussion to the related work.
>
> 2. It seems a bit odd that the results are only on the proposed new dataset. Shouldn't the paper include standard existing benchmarks for the task?
>
> Thank you for referring to the paper. We also discuss several other works that perform token-level segmentation on prompts in related work in Lines 151-153. However, one problem with these works discussed in related work, and the mentioned one is that they require the model to generate a sequence of tokens with the same length as the input, which can cost a lot when the input sequence is long. To address this, we propose a new training formulation that the model learns to generate only starting tokens of each sequence as output, which is described in Sec 4.1. We will add this discussion to the related work.
>
> |                       | recon_rate | em        | pk       | label_f1  | character_f1 | Avg       |
> |-----------------------|---------------------|-----------|----------|-----------|--------------|-----------|
> | SFT                   | 99.3               | 70.1     | 6.6     | 81.7     | 81.6        | 85.2     |
> | SFT w/2epoches        | 99.3               | **71.3** | 6.4     | 81.9     | 82.5        | 85.7     |
> | SFT+RLVR              | 99.4               | 71.1     | 6.3     | **82.6** | 82.4        | 85.9     |
> | SFT+RLVR w/high temp. | **99.5**           | 70.9     | 6.4     | 82.4     | 82.4        | 85.7     |
> | RL-PLUS               | 99.3               | 70.9     | 6.3     | 81.9     | 82.5        | 85.7     |
> | BoundRL               | 99.4               | 71.2     | **6.2** | 82.5     | **82.7**    | **85.9** |
>
> 3. It's a little hard to tell from Table 3 that the baselines represent the state-of-the-art. RL-PLUS is the only meaningful third-party baseline, and others are just in-house ablations, it seems? For someone who doesn't have a background on recent text seg research, it's not clear how strong the performance win is.
>
> In addition to the mentioned baselines, we also include a few-shot baseline using Claude3.5 as in PROMPTPRISM [1]. We also include a NER baseline, which is commonly used for token classification. As shown in Table 5, our proposed BoundRL outperforms these baselines. We do not include previous text segmentation as most of them are designed for sentence-level or paragraph-level segmentation and are not applicable for token-level segmentation, as discussed in Lines 146-150.
>
> [1] PROMPTPRISM: A Linguistically-Inspired Taxonomy for Prompts

---

> > ### Author Response · Authors · 2025-12-02
> >
> > To further show why we do not compare BoundRL with previous works using sentence-level segmentation, we develop a new sentence-level oracle baseline, Oracle-sent, based on the ground-truth annotation of segments. Specifically, we first segment the input texts into sentences using nltk.sent_tokenize, and then label each sentence with the label that has the maximum overlap. We then merge neighboring sentences with the label into a single segment. We then report the performance of  Oracle-sent in the following table.
> >
> > |             |  synthetic |      |       |          |              |  langchain |      |       |          |              |       |
> > |-------------|:----------:|:----:|:-----:|:--------:|:------------:|:----------:|:----:|:-----:|:--------:|:------------:|-------|
> > |             | recon_rate | em   | pk    | label_f1 | character_f1 | recon_rate | em   | pk    | label_f1 | character_f1 | Avg.  |
> > | Oracle-sent | 100.0      | 2.7  | 16.5  | 99.7     | 90.3         | 100.0      | 5.2  | 18.9  | 99.5     | 92.2         | 75.4  |
> >
> > From the table, we can observe that BoundRL outperforms Oracle-sent even though Oracle-sent directly uses the ground-truth annotation. The result shows that sentence-level segmentation methods cannot work well on our dataset and we only include token-level segmentation methods as baselines.

---

### Official Review · Reviewer_8hZx · 2025-10-31

**Soundness:** 1
**Presentation:** 3
**Contribution:** 1
**Rating:** 2
**Confidence:** 4

**Summary:**

The paper applies a variant of RLVR to boundary detection in language model prompts. It shows that its method of only generating tokens that indicate boundaries is faster than using a prompt-based approach.

**Strengths:**

In looking at text segmentation, the paper picks up a topic that nowadays receives less attention and may by many already (falsely) considered as solved. It focused on issues with blindly applying LLMs to the problem and points to an alternative that improves over that approach.

**Weaknesses:**

At a high level, this paper feels like an engineering project that was overly quickly turned into the format of an academic paper. I am lacking rigor in baselines and proper justification for why many of the choices that were made.

**Why RL**

This paper presents an RLVR hammer and then turns text segmentation into a nail. The paper does not discuss why it is necessary to use RL for this approach. Even after reading it, I am left wondering if it is necessary to treat this as a text generation problem and why I should care about this approach at all. The related work at minimum should outline this argument.

**Lack of Baselines and Datasets**

The probably strongest weakness of this paper is that it evaluates prompt text segmentation via a very limited set of experiments. There are a myriad classic text segmentation benchmarks with strong baselines for scientific papers, corporate filings, emails, and many others. There is no justification given beyond "prompts are complex".

6.2 is discussing how efficient BoundRL is but it is generating a significant 119 tokens when a classic sequence tagging approach would not have to generate any of this. While the NER baseline is discussed as scoring lower and achieving only fragmented, short segments, this is in disagreement with the entire body of literature. I strongly suspect a faulty implementation. For, example, the NER baseline uses an autoregressive model. This has historically not worked as well as dedicated sequence labeling setups. But, due to the lack of baselines, it is impossible to assess this.

Even if no classic models are explored, I would have expected baselines based on LayoutLM and similar models aimed to improve document understanding.

**Lack of humans in the process**

The paper relies to a large degree on synthetic data. The only test on human-created prompts is via the langchain hub. But this dataset is not described or analyzed and there is no human evaluation of the results. Given that the langchain hub has templates, it will be biased in many ways. In addition, the distribution over topics is very top-heavy with a focus on certain types of agents over others.

**Questions:**

n/a

---

> ### Author Response · Authors · 2025-11-25
>
> Thank you for your review.
>
> 1. Why treat this as a text generation problem?
>
> As discussed in the related work (Lines 144-146), our project is not the first project that treats text segmentation as a generation task and there are already several previous works treat the text segmentation as generation tasks [1,2]  However, these approaches perform segmentation on the sentence or paragraph level, which face limitations when applied to structured texts with code snippets, structured data formats (JSON, XML) that do not follow boundary patterns of traditional sentence or paragraph.
>
> [1] Structured Summarization: Unified Text Segmentation and Segment Labeling as a Generation Task
>
> [2] Lumberchunker: Long-form narrative document segmentation
>
> 2. Why it is necessary to use RL for this approach?
>
> We discuss why we need to perform RL in addition to SFT in the introduction (Lines 83-87). SFT can mistakenly penalize starting tokens that correspond to the right boundary positions and provide insufficient penalties for minor token mismatches that cause failures during segment parsing, e.g., locating starting tokens. Contrarily, RL can address these issues with carefully designed rewards.
>
> 3. The probably strongest weakness of this paper is that it evaluates prompt text segmentation via a very limited set of experiments. There are a myriad classic text segmentation benchmarks with strong baselines for scientific papers, corporate filings, emails, and many others. There is no justification given beyond "prompts are complex".
>
> We are aware of the fact that there are already text segmentation benchmarks in different domains. However, one issue with these benchmarks is that they are plain texts and can be segmented on the sentence or paragraph level. Contrarily, prompts contain code snippets, structured data formats (JSON, XML), and placeholders that do not follow the boundary patterns of traditional sentences or paragraphs. Therefore, prompt segmentation can only be performed on the token level.
>
> 4. While the NER baseline is discussed as scoring lower and achieving only fragmented, short segments, this is in disagreement with the entire body of literature. I strongly suspect a faulty implementation. For example, the NER baseline uses an autoregressive model. This has historically not worked as well as dedicated sequence labeling setups. But, due to the lack of baselines, it is impossible to assess this.
>
> To implement the NER baseline, we use the code from https://huggingface.co/docs/transformers/tasks/token_classification. Besides, we perform a hyperparameter search for learning rate and training epoch to improve the performance of the NER baseline. Furthermore, for NER baselines in Table 3, it can be observed that they generally perform well on Character-F1, which they are directly optimized for. If they were indeed falsely implemented, they would show bad performance on Character-F1.
>
> Following your request, we additionally finetune ModernBERT-large, an SOTA pretrained bi-directional encoder on our training dataset. We finetune the model for two epochs with a learning rate of 2e-5.  The results are shown in the following table.
>
>
> |     | Synthetic           |       |      |          |              | Langchain           |       |       |          |              |       |
> |-----|---------------------|-------|------|----------|--------------|---------------------|-------|-------|----------|--------------|-------|
> |     | recon_rate | em    | pk   | label_f1 | character_f1 | recon_rate | em    | pk    | label_f1 | character_f1 | Avg.  |
> | NER | 99.7                | 57.3  | 5.7  | 85.8     | 95.2         | 98.8                | 19.4  | 15.6  | 66.4     | 86.5         | 78.8  |
>
> Compared with the results in Table 5, NER using ModernBERT-large outperforms NER using autoregressive models but still underperforms other baselines. Another takeaway is that the issue of fragmented segments became less severe but still exists.
>
> 5. Langchain dataset is not described or analyzed.
>
> We describe the statistics of the dataset in Table 1 and analyze the proportion of components in Figure 8 (appendix). We will show more detailed examples of annotated Langchain prompts in the appendix in the revision.
>
> 6. There is no human evaluation of the results.
>
> All prompts in the langchain and synthetic subset are annotated by a group of trained annotators with detailed guidelines (Lines 296-301), and extensive quality assurance (programmatic and human inspection) was done to filter out low quality annotations. Therefore, the human annotation of each prompt is of high quality. We will include more details about annotation guidelines and QA protocols in the appendix in the revision.

---

> > ### Author Response · Authors · 2025-11-25
> >
> > 7. The distribution over topics is very top-heavy with a focus on certain types of agents over others.
> >
> > We agree that the distribution of labels is not balanced, however this is expected and indeed reflected in Table 3 of a comprehensive analysis across LLM prompts from LLMApp repos [1]. Therefore, we evaluate the quality of generated segmentation using Label-F1 and Character-F1 instead of accuracy.
> >
> > [1] From Prompts to Templates: A Systematic Prompt Template Analysis for Real-world LLMapps
> >
> > 8. Given that the langchain hub has templates, it will be biased in many ways.
> >
> >
> > The langchain hub templates that were used for our experiments were carefully selected so that they meet the complexity and diversity criteria, covering a variety of LLM tasks that possess different characteristics in the prompt templates, such as formatting, the ways of providing few-shot examples, tool specifications, output format restrictions, etc. Furthermore, the synthetic prompt data used for training covers even a broader range of task types, writing styles, format types, prompt length, level of details, etc., to ensure diversity and avoid bias towards certain domains and styles. Therefore, we consider that both the data used for training and benchmarking are unbiased.

---

> > > ### Author Response · Authors · 2025-12-02
> > >
> > > To further show why we do not compare BoundRL with previous works using sentence-level segmentation, we develop a new sentence-level oracle baseline, Oracle-sent, based on the ground-truth annotation of segments. Specifically, we first segment the input texts into sentences using nltk.sent_tokenize, and then label each sentence with the label that has the maximum overlap. We then merge neighboring sentences with the label into a single segment. We then report the performance of  Oracle-sent in the following table.
> > >
> > > |             |  synthetic |      |       |          |              |  langchain |      |       |          |              |       |
> > > |-------------|:----------:|:----:|:-----:|:--------:|:------------:|:----------:|:----:|:-----:|:--------:|:------------:|-------|
> > > |             | recon_rate | em   | pk    | label_f1 | character_f1 | recon_rate | em   | pk    | label_f1 | character_f1 | Avg.  |
> > > | Oracle-sent | 100.0      | 2.7  | 16.5  | 99.7     | 90.3         | 100.0      | 5.2  | 18.9  | 99.5     | 92.2         | 75.4  |
> > >
> > > From the table, we can observe that BoundRL outperforms Oracle-sent even though Oracle-sent directly uses the ground-truth annotation. The result shows that sentence-level segmentation methods cannot work well on our dataset and we only include token-level segmentation methods as baselines.

---

### Official Review · Reviewer_1KdK · 2025-11-01

**Soundness:** 2
**Presentation:** 3
**Contribution:** 3
**Rating:** 4
**Confidence:** 2

**Summary:**

This paper proposes a method for text segmentation of structured text that reduces the computational load on models as well as improving performance. The authors frame the problem as boundary identification and task a model to identify where various roles start and end throughout the text, assigning these segments the appropriate label. They then use these labels and segments to segment the entire text. They observe that by using this method they are able to improve the efficiency of text segmentation. They also observe that their method is able to segment text more accurately than few-shot prompted Claude models.

**Strengths:**

The problem is well described, and the paper is clearly written. The problem is relevant. The method improves significantly in terms of efficiency and does improve performance when compared to Claude models.

**Weaknesses:**

1. While the improvements show in table 3 are significant, the method improves over Claude inconsistently across metrics. It would be good to clarify why this might be and offer some explanation as to which metrics are the most important.

2. While there are lots of experimental results comparing different settings of BoundRL and different fine-tuning paradigms, and there are experimental results comparing to few-shot prompting of Claude, the performance is not compared to methods from prior work. It would be good to see how both performance and efficiency compares to these methods.

**Questions:**

1. Do you have any insights for the differences in metric improvement in Table 3?

2. Why do you choose few-shot prompting of Claude to compare against vs other prior work?

**Details Of Ethics Concerns:**

Annotators are used, but I see no discussion of the rate they were paid.

---

> ### Author Response · Authors · 2025-11-25
>
> Thank you for your review.
>
> 1. While the improvements show in table 3 are significant, the method improves over Claude inconsistently across metrics. It would be good to clarify why this might be and offer some explanation as to which metrics are the most important.
>
> The inconsistency in improvement across metrics is because different metrics are measuring different aspects of the segmentation effectiveness, and they are all important. For example,  a small mismatch in any character (e.g., a space or a new line character) can lead to a substantial drop in exact match since it is the most strict measure, but has a relatively smaller impact on character F1. Reconstruction ratio measures to what extent the prompt can be reconstructed, ensuring important information is not missed, Pk measures the boundary position, Label F1 measures the label accuracy, and exact match and character F1 measures both boundary position and label accuracy.
>
> 2. the performance is not compared to methods from prior work. It would be good to see how both performance and efficiency compares to these methods.
>
> As we discussed in related work, most previous works on text segmentation perform segmentation on the sentence or paragraph level, which face limitations when applied to structured texts with code snippets, structured data formats (JSON, XML) that do not follow the boundary patterns of traditional sentences or paragraphs, and this is especially important for GenAI prompts. Therefore, we only include these baselines that are applicable for token-level segmentations.
>
> 3. Do you have any insights for the differences in metric improvement in Table 3?
>
> We believe the improvements between SFT and RLVR can be attributed to two factors. First, SFT can mistakenly penalize starting tokens that correspond to the right boundary positions and provide insufficient penalties for minor token mismatches that cause failures in locating starting tokens. For the improvements between BoundRL and RLVR, we believe it is because BoundRL addresses the entropy collapse issue of RLVR  by perturbing a fraction of generated sequences of segments through boundary adjustments and label modifications.
>
> 4. Why do you choose few-shot prompting of Claude to compare against vs other prior work?
>
> We use Claude following the previous practice of PromptPrism [1]. We use a few-shot prompting to help the model better understand the task and expected output format. Even with few-shot prompting, we demonstrated that our method outperforms proprietary LLMs.
>
> [1] PROMPTPRISM: A Linguistically-Inspired Taxonomy for Prompts

---

> > ### Author Response · Authors · 2025-12-02
> >
> > To further show why we do not compare BoundRL with previous works using sentence-level segmentation, we develop a new sentence-level oracle baseline, Oracle-sent, based on the ground-truth annotation of segments. Specifically, we first segment the input texts into sentences using nltk.sent_tokenize, and then label each sentence with the label that has the maximum overlap. We then merge neighboring sentences with the label into a single segment. We then report the performance of  Oracle-sent in the following table.
> >
> > |             |  synthetic |      |       |          |              |  langchain |      |       |          |              |       |
> > |-------------|:----------:|:----:|:-----:|:--------:|:------------:|:----------:|:----:|:-----:|:--------:|:------------:|-------|
> > |             | recon_rate | em   | pk    | label_f1 | character_f1 | recon_rate | em   | pk    | label_f1 | character_f1 | Avg.  |
> > | Oracle-sent | 100.0      | 2.7  | 16.5  | 99.7     | 90.3         | 100.0      | 5.2  | 18.9  | 99.5     | 92.2         | 75.4  |
> >
> > From the table, we can observe that BoundRL outperforms Oracle-sent even though Oracle-sent directly uses the ground-truth annotation. The result shows that sentence-level segmentation methods cannot work well on our dataset and we only include token-level segmentation methods as baselines.

---

### Note · Authors · 2026-01-06

I have read and agree with the venue's withdrawal policy on behalf of myself and my co-authors.